# Taphonomic information from the modern vertebrate death assemblage of Doñana National Park, Spain

M. Soledad Domingo[1¤]*, David M. Martín-Perea[2,3,4], Catherine Badgley[5], Enrique Cantero[2], Paloma López-Guerrero[3], Adriana Oliver[6], Juan José Negro[1]

1 Departamento de Ecología Evolutiva, Estación Biológica de Doñana-CSIC, Seville, Spain, 2 Departamento de Paleobiología, Museo Nacional de Ciencias Naturales-CSIC, Madrid, Spain, 3 Departamento de Geodinámica, Estratigrafía y Paleontología, Facultad de Ciencias Geológicas, Universidad Complutense de Madrid, Madrid, Spain, 4 Institute of Human Evolution in Africa – IDEA, Madrid, Spain, 5 Department of Ecology and Evolutionary Biology, University of Michigan, Ann Arbor, Michigan, United States of America, 6 Asociación Mujeres con los Pies en la Tierra, Madrid, Spain

¤ Current address: Departamento de Didáctica de las Ciencias Experimentales, Ciencias Sociales y Matemáticas, Facultad de Educación, Universidad Complutense de Madrid, Madrid, Spain
* mariasod@ucm.es

**Data Availability Statement:** All relevant data are within the manuscript.

**Funding:** MSD received funding for this project from the European Union's Horizon 2020 research

## Abstract

Modern death assemblages provide insights about the early stages of fossilization and useful ecological information about the species inhabiting the ecosystem. We present the results of taphonomic monitoring of modern vertebrate carcasses and bones from Doñana National Park, a Mediterranean coastal ecosystem in Andalusia, Spain. Ten different habitats were surveyed. Half of them occur in active depositional environments (marshland, lake margin, river margin, beach and dunes). Most of the skeletal remains belong to land mammals larger than 5 kg in body weight (mainly wild and feral ungulates). Overall, the Doñana bone assemblage shows good preservation with little damage to the bones, partly as a consequence of the low predator pressure on large vertebrates. Assemblages from active depositional habitats differ significantly from other habitats in terms of the higher incidence of breakage and chewing marks on bones in the latter, which result from scavenging, mainly by wild boar and red fox. The lake-margin and river-margin death assemblages have high concentrations of well preserved bones that are undergoing burial and offer the greatest potential to produce fossil assemblages. The spatial distribution of species in the Doñana death assemblage generally reflects the preferred habitats of the species in life. Meadows seem to be a preferred winter habitat for male deer, given the high number of shed antlers recorded there. This study is further proof that taphonomy can provide powerful insights to better understand the ecology of modern species and to infer past and future scenarios for the fossil record.

## Introduction

The fossil record is our primary window to study life of the past and to infer episodes of faunal and floral change associated with past environmental changes. Nevertheless, before obtaining

and innovation programme under grant agreement MSCA-700196 (Marie Skłodowska-Curie Individual Fellowships programme). The funders had no role in study design, data collection and analysis, decision to publish, or preparation of the manuscript.

Competing interests: The authors have declared that no competing interests exist.

evolutionary or environmental information from the fossil record, it is necessary to evaluate the taphonomic processes that have transformed information from living faunal and floral communities to death assemblages and eventually to fossil assemblages [1–3]. A thorough understanding of the processes that result in fossilization can only be achieved if taphonomic analyses are also conducted in death assemblages from modern ecosystems (i.e., actualistic studies). Kidwell and Tomasovych [4] defined a death assemblage as the dead or discarded organic remains that are encountered in still largely unburied form on landscapes and sea-floors, and thus are distinct from the permanently buried fossil record. Death-assemblage surveys in modern ecosystems have documented comprehensive information about the multiple processes affecting organic remains and controlling their eventual recycling or preservation [e.g., 2, 5].

In the field of vertebrate taphonomy, Johannes Weigelt [6] conducted one of the first studies of the processes governing the mortality, decomposition, and potential preservation of modern vertebrate carcasses. In the 1970s, Anna K. Behrensmeyer started a long-term mammal bone survey in Amboseli National Park (Kenya) that constitutes a paradigm for this type of analysis [7,8]. For almost 50 years, Behrensmeyer and colleagues have monitored carcasses of medium and large vertebrates in Amboseli National Park in order to compare the faunal composition and abundance (fidelity) in the bone assemblage to those of the living community and to document the post-mortem processes that modify modern bone assemblages and result in either preservation or destruction [8,9]. In addition to offering important insights about the taphonomic modifications that affect recent death assemblages, these studies have shown that bone surveys are also a useful, non-invasive way to track aspects of living populations (e.g., habitat and resource utilization, mortality through time, or changes in faunal abundance, including extirpation of species from the study area), and thereby can inform decisions about conservation biology and wildlife management [e.g., 8, 10–15]. Since these initial efforts, additional vertebrate bone surveys have been conducted. Most of them have been carried out in regions with tropical or temperate climatic regimes, and mainly are located in Africa and the New World [e.g., 5, 8, 11–26].

We seek to expand the taphonomic monitoring of modern vertebrate remains to Mediterranean ecosystems in order to provide baseline information for understanding the processes leading to fossilization of terrestrial vertebrates in a Mediterranean climatic regime. The "Mediterranean" climate is characterized by mild, wet winters and hot, dry summers. These ecosystems typically occur on the west side of continents between about 30˚ and 40˚ latitude, although the reference area lies in the regions surrounding the Mediterranean Sea [27]. Several studies, mainly focused on pollen data, have placed the onset of the Mediterranean-type climate at around 3.4 Ma [28,29]. Tzedakis [30] questioned this age in a review article and proposed that the Mediterranean climatic regime was present earlier, at least intermittently, during the course of the Cenozoic (or even earlier). This author mentioned several geological and paleontological lines of evidence that would demonstrate the early, discontinuous occurrence of a Mediterranean-type climatic regime, such as the tree-ring pattern displayed by fossil conifers in the Late Jurassic at 36˚ paleolatitude; the presence of evaporite, phosphorite and coal deposits at 25˚-30˚ paleolatitudes during the Maastrichtian; the abundance of soil carbonates in paleosols from Greece and Turkey over the past 11 Myr; and the sustained dominance of $C_3$ plants (and absence of $C_4$ vegetation) in the Mediterranean Basin over the last 11 Myr (pointing to reduced summer precipitation typical of the Mediterranean climate) [30].

Our study site is Doñana National Park (DNP), a UNESCO World Heritage Site and Biosphere Reserve, in a Mediterranean biome in Andalusia, Spain (Fig 1). DNP is an excellent natural laboratory for actualistic taphonomic research for several reasons. First, it has been protected as a natural area since 1969 and activities are restricted to conservation and research

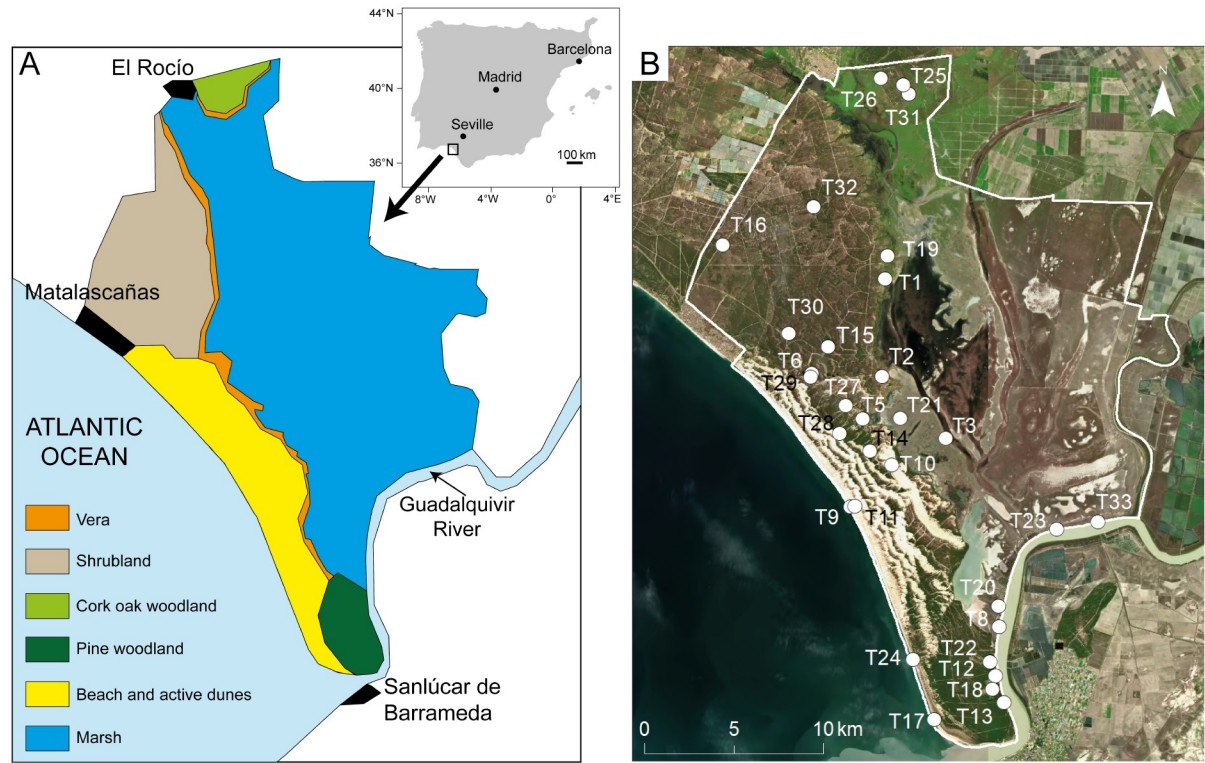

**Fig 1. Geographic location of Doñana National Park and transects sampled.** (A) Distribution of the major habitats of Doñana National Park. The upper right inset shows the location of the park in the Iberian Peninsula. (B) Orthophotomap showing the location of the transects sampled in this study (OrtoPNOA 2019 CC-BY 4.0 scne.es). T = transect.

over much of the area (tourism takes place in the periphery of the Park and only limited guided tours are permitted inside the Park). Second, this natural area has been intensively studied with several decades of records of the physical environment and systematic monitoring of selected vertebrate species. Third, it contains several habitats that follow the local geomorphology. Fourth, DNP has a rich vertebrate fauna. DNP constitutes a sanctuary for birds that migrate annually from Africa to Europe and vice versa. Many of these species stay at DNP during the summer or the winter. Many other bird species reside permanently in DNP. In addition, DNP is home to two critically endangered Iberian species, the Iberian Lynx (*Lynx pardina*) and the Spanish Imperial Eagle (*Aquila adalberti*). The non-marine vertebrate fauna includes 20 species of freshwater fishes, 11 amphibians, 21 reptiles, 37 mammals, and 360 birds [31].

DNP landscapes include active dunes, shrubland and woodlands developed on stabilised dunes, marshland, and an ecotone (called 'Vera') in the boundary between the stabilised dunes and the marshes (Figs 1 and 2, Table 1). Several active depositional settings with fossiliferous analogues in the geological record occur (e.g., river channels, floodplains, lake bed/margin, dunes, and beaches).

DNP has a Mediterranean subhumid climate with Atlantic influence [36,37]. The average annual rainfall is 580 mm with substantial interannual variation, ranging from 170 mm (2004–2005) to 1000 mm (1995–1996) [36]. Generally, Doñana has well-defined seasonal precipitation, with a wet season between October and April and a dry season between May and September. The mean annual temperature is 17˚C [36]. In the winter, temperatures are usually mild (generally above freezing). Summers are long and hot (>35˚C occurs frequently).

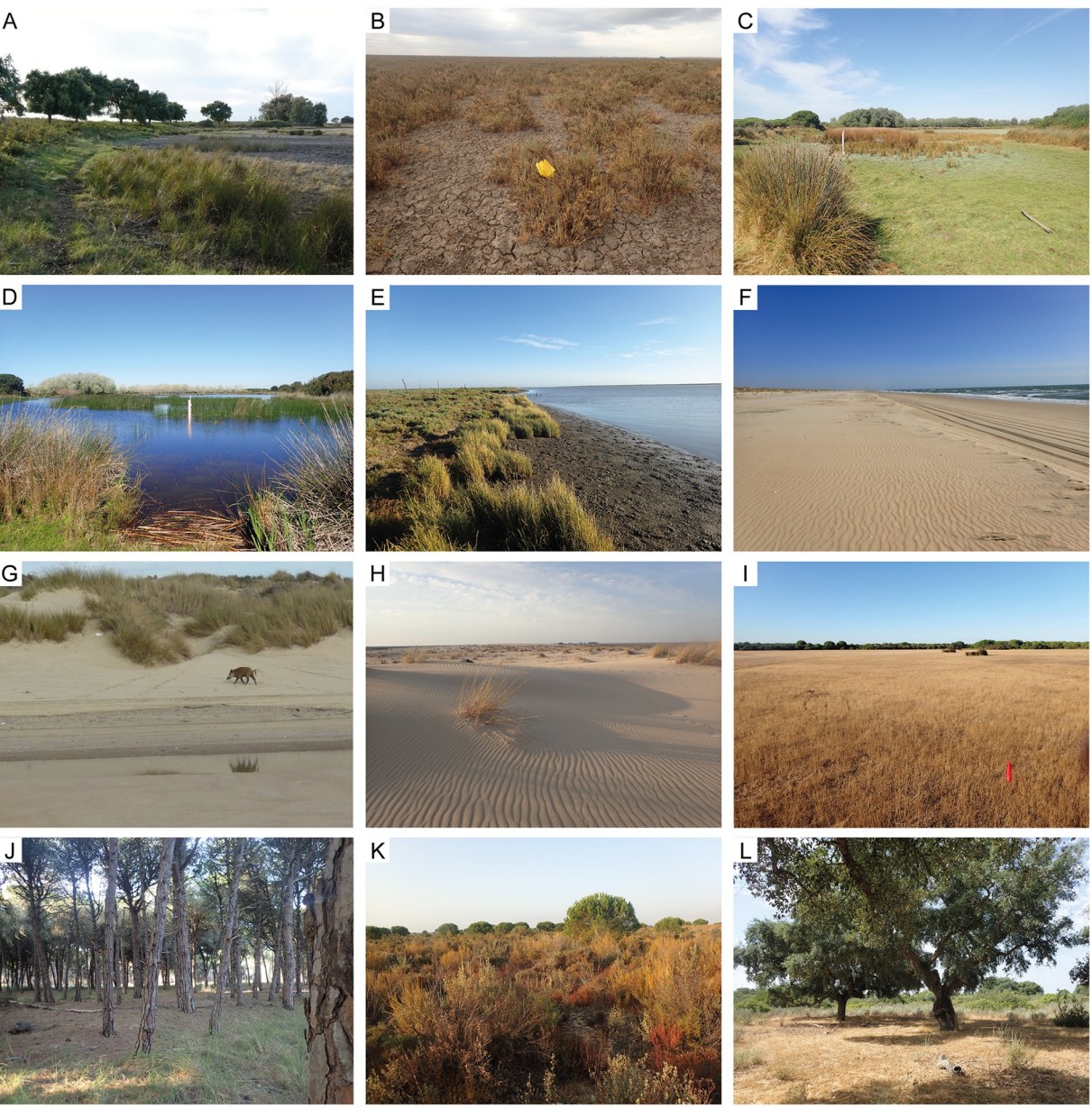

**Fig 2. Major habitats of Doñana National Park.** (A) Boundary between the Vera ecotone (left, line of trees) and the marsh (right). (B) Desiccated marshland (September, 2018). (C) Lake-margin/bed habitat. Desiccated Sopetón Lake (September, 2017). (D) Sopetón Lake with water (March, 2018). (E) Guadalquivir River margin. (F) Beach. (G) A wild boar roaming on the beach, where the field of dunes starts. (H) Active dunes (Greylag Geese Hill). (I) Meadow habitat in Las Marismillas area. (J) Pine-woodland habitat. (K) Shrubland habitat. (L) Cork-oak woodland.

In this study, we focus on the biostratinomic (= before burial) modifications of bones from DNP, highlighting the similarities and differences observed among the different habitats sampled and establishing, where possible, relationships with taphonomic features of the vertebrate fossil record. We present the species composition of mammals from each habitat based on the bones and carcasses found and qualitatively compare them to the habitat preferences of the living mammals. Quantitative studies of the concordance (fidelity) between the composition and abundance of the death assemblage to the live censuses will be undertaken in the future.

**Table 1. Description of major habitats in Doñana National Park.**

| Habitat | Description |
|---|---|
| Beach | This habitat is 30 km long and extends from the town of Matalascañas to the mouth of the Guadalquivir River. The width ranges from 100 to 300 m. The beach occurs on sand bars that developed in this area after the Flandrian transgression (ca. 6500 years BP) [32]. |
| Active dunes | The dunes are parallel to the Atlantic shoreline and constitute the largest active dune field in Europe [32]. They are formed by up to five fronts alternating with low areas (slacks) where pine woods (*Pinus pinea*) and junipers (*Juniperus macrocarpa*) grow. |
| Shrubland and woodland on stabilized dunes | These areas are known as "Cotos" or "Monte." The sand dunes here are stabilized by vegetation. The vegetation distribution in this habitat is variable and depends on water availability, which is determined by the depth of the water table. The shrubland contains over 30 species of woody bushes. The woodland mainly consists of pines, junipers (*Juniper phoenicia sub. turbinata*) and cork oaks (*Quercus suber*). A number of phreatic lakes occur in this habitat. Some of these lakes are virtually permanent, whereas others are seasonal. Our lake-margin transects were carried out along the permanent lakes of Santa Olalla and Sopetón [33]. In the southern edge of the park, in the area known as "Las Marismillas," there are openings of pasture among dense pine woods denoted in this work as 'meadows.' |
| Vera | This name is given to the ecotone between the shrubland and woodland in the stabilised dunes and the marshlands [31]. It forms a long, narrow fringe whose width ranges from several hundred meters to only a few meters, even disappearing in some areas. It constitutes the boundary between sandy and clayey soils, where the moisture filters through the sandy layers, favoring the development of pastures and reeds. Large, isolated cork oaks dot the Vera, although many have recently died from an exotic mould (*Phytophthora cinnamomi*). |
| Marshland | Doñana marshes are one of the largest freshwater wetlands in Europe, covering 34,000 ha [33]. This is the most extensive habitat in DNP. It is a vast muddy plain that floods seasonally. The Doñana marshland is fed primarily by rainfall and several drainages [34]. Secondarily, it is fed by groundwater discharge along the margin and through spring-fed streams. The tidal influence from the estuary of the Guadalquivir River is minimal at present, following the construction of a dyke parallel to the Guadalquivir River in 1984 and its enlargement after the Aznalcóllar mining spill in 1998 [35]. The maximum inundation level of the marshes typically occurs in February. The average water depth in the marshes is about 0.5 meters [36]. At the end of July, the marshes are nearly dry. The marshes constitute a sanctuary for birds that migrate annually from Africa to Europe and vice versa. |

## Previous taphonomic work at Doñana National Park

Eloísa Bernáldez Sánchez and co-workers sampled carcasses in Doñana National Park between 1988 and 1991 with the aim of understanding the patterns that lead to carcass degradation and the influence of factors such as body size, time elapsed since death, and vegetation coverage [38,39]. They restricted their study to the Doñana Biological Reserve, a relatively small area (6,900 ha) within DNP, lacking some major habitats of the park (e.g., beach and river margin). By mainly focusing on the monitoring of complete or nearly complete carcasses, their studies differed methodologically from ours (i.e., we have also analyzed many isolated bones). The main variables that they recorded were measures of skeletal disarticulation, bone dispersal, and bone loss [38,39], whereas our observations include a larger set of taphonomic variables and surveys over the full suite of DNP habitats. We consider therefore that their study and our study are not redundant but complementary.

## Material and methods

Permits for conducting field work at Doñana National Park were granted by the Director of Espacio Natural de Doñana (Project number 2017/07). We established a total of 31 transects distributed over ten habitats as follows: Vera ecotone (3 transects), marshland (3 transects), lake-bed/margin (3 transects), river margin (3 transects), beach (3 transects), active dunes (3 transects), meadow (3 transects), pine woodland (3 transects), shrubland (4 transects), and cork-oak woodland (3 transects) (Figs 1B and 2). Among these habitats, the marshland, lake-bed/margin, river margin, beach and active dunes are active depositional systems (sediment accumulation rates are higher than in the remaining habitats). Field work took place for five weeks in September, 2017 (18 transects) and in September, 2018 (13 transects). This period coincided with the end of the dry season, when the marshes were dry and could be easily traversed and sampled. The information from the two field seasons is combined in this study.

Transects were 1 km long, but we terminated the survey upon reaching 20 sampled individuals (following [5]). To accomplish the surveys, we followed different approaches depending on visibility conditions. In habitats with abundant vegetation, we set flags along the midline of the transect and walked back and forth perpendicular to this line, flagging all the bones, antlers, and carcasses that we encountered. If vegetation was absent or low and scattered, we did not flag the midline but instead projected the 1 km length and orientation of the transect using a GPS unit (GARMIN GPSMAP 64s) in the direction we wanted to follow. One of us walked the midline using the GPS for orientation, while the other members of the team walked parallel to one another and at a constant distance apart on one side of the midline. Once we reached 1 km, we walked back on the other side of the midline in a similar manner. In the lake, river and beach habitats, we walked parallel to the water's edge.

In habitats where visibility was good, we extended the transects 50 m to each side of the midline so the total width of the plot was 100 m. We covered 30 m to each side of the center line if visibility was limited (60 m total width). Transects were set well apart from each other (the minimum distance between two transects was 500 m, but usually the distance was greater than 1 km), so that remains from one individual were not likely to be sampled in more than one transect. The survey process described here is suitable for documenting terrestrial vertebrates larger than 5 kg in body weight (although we noted bones from smaller species as well). In DNP, such vertebrates include mammals, mainly wild and feral ungulates, and several bird species (Table 2).

Taphonomic information for each bone was recorded on paper data sheets and included variables traditionally recorded in studies of vertebrate taphonomy: skeletal-element identification, species identification, geographic location, degree of articulation (articulated, associated or isolated), right or left side, ontogenetic age (infant, juvenile, adult, old), element completeness, degree of burial, weathering stage, abrasion stage, and quantification and description of trampling or chewing marks. Element completeness was assessed by determining the integrity of each identifiable element using the following categories (CS = Completeness Stage): complete (CS 0), almost complete (CS 1; bones only missing a bone chip), more than one-half complete (CS 2), and one-half complete or less (CS 3) [45]. Weathering measures the deterioration of bone mainly due to physical and chemical agents related to local environmental factors and is proportional to the time elapsed since the death of the animal [46]. Weathering evaluation followed the scale of Behrensmeyer [46] that ranges from 0 (unweathered) to 5 (extremely weathered). Each stage corresponds to the number of years since death (WS = Weathering Stage): WS 0 (0–1 year), WS 1 (6 months–2.5 years), WS 2 (2–4 years), WS 3 (4–8 years), WS 4 (6.5–20 years), WS 5 (10 to >25 years). Years must be interpreted with caution, as they were estimated for bones from Amboseli National Park, a tropical

**Table 2. List of mammals and birds larger than 5 kg body mass from Doñana National Park.**

| MAMMALS | | | |
|---|---|---|---|
| **Order/Family** | **Species** | **Body mass (kg)** | **Relative Abundance** |
| Artiodactyla/Bovidae | Cattle (Bos taurus)* | 240.0–750.0 | Abundant |
| Perissodactyla/Equidae | Feral horse (Equus caballus)* | > 300.0 | Abundant |
| Artiodactyla/Cervidae | Red deer (Cervus elaphus)* | 100.0–150.0 | Abundant |
| Artiodactyla/Bovidae | Sheep (Ovis aries)* | 45–100 | Occasional |
| Artiodactyla/Suidae | Wild boar (Sus scrofa)* | 55.0–85.0 | Abundant |
| Artiodactyla/Cervidae | Fallow deer (Dama dama)* | 28.0–63.0 | Abundant |
| Carnivora/Canidae | Gray wolf (Canis lupus) | 20.0–40.0 | Extinct since 1952 |
| Carnivora/Felidae | Iberian lynx (Lynx pardinus) | 6.0–15.0 | Rare (endangered) |
| Carnivora/Mustelidae | European otter (Lutra lutra) | 6.5–10.0 | Common |
| Carnivora/Mustelidae | European badger (Meles meles) | 4.8–9.3 | Common |
| Carnivora/Canidae | Red fox (Vulpes vulpes)* | 3.0–8.0 | Abundant |
| Carnivora/Felidae | Wildcat (Felis sylvestris) | 3.0–5.0 | Rare |
| BIRDS | | | |
| **Order/Family** | **Species** | **Body mass (kg)** | **Relative Abundance** |
| Otidiformes/Otididae | Great bustard (Otis tarda) | 3.1–18 | Occasional |
| Accipitriformes/Accipitridae | Cinereus vulture (Aegypius monachus) | 6.8–14.0 | Rare |
| Accipitriformes/Accipitridae | Griffon vulture (Gyps fulvus) | 6.2–11.3 | Abundant |
| Gruiformes/Gruidae | Common crane (Grus grus) | 3.9–7.0 | Abundant |
| Anseriformes/Anatidae | Greylag goose (Anser anser) | 2.2–5.0 | Abundant |

Species are listed from larger to smaller body mass. The asterisks denote species identified in the bone sample (we also identified remains of the Egyptian mongoose, *Herpestes ichneumon*, but this species is not included in this table because its body mass is less than 5 kg). Data for mammals are from [40,41]. Data for birds are from [42–44].

savannah with environmental conditions different from those of DNP. In any event, weathering stages can still be used for comparative purposes. We are currently establishing our own weathering-stage scale with bones that we monitor annually at DNP.

Abrasion refers to the erosion on bones caused by the impacts of sedimentary particles and results in the smoothing and polishing of bones. Abrasion was scored in three stages following Alcalá [47] (AS = Abrasion Stage): intact bone (AS 1), moderately rounded bone (AS 2), and polished bone (AS 3). Tooth (chewing) marks were classified as pits (shallow oval depression), punctures (deep oval depression), scores (elongated marks), and furrows (contiguous grooves, generally located at the ends of long bones) [48]. Chewed antlers from the two deer species in the park were also documented.

To assess the anatomical and taxonomic composition of skeletal remains in the transects, we used the NISP (= Number of Identified Specimens) index. A specimen is considered to be a bone, a tooth, or a fragment of either [49]. We estimated the Minimum Number of Individuals (MNI) from each transect based on the species identification, skeletal element, anatomical side, age, and weathering stage. We followed the recommendation of Behrensmeyer and Miller [5] of assuming that an isolated bone did not represent a new individual unless it displayed a feature pointing to the contrary. During our survey, most of the bones were left in the field. Bones with an uncertain identity or displaying interesting taphonomic features were collected for further analysis.

In order to characterize the habitats as a function of their biostratinomic modifications, and to facilitate the evaluation of similarities and differences among them, we performed a

Principal Component Analysis (PCA) of taphonomic features. Taphonomic variables used in the PCA were: %CS2+CS3, %WS≥3, %Abraded bones, %Trampled bones, %Buried bones, %Chewed bones, and %Isolated bones for each habitat. We did not use the complementary variables to those listed (i.e., %CS0+CS1, %WS0-2, %Non-abraded bones, %Non-trampled bones, %Non-buried bones, %Non-chewed bones, and %Articulated + Associated bones), but as we used percentages these variables are already implicitly present in the analysis. We used the covariance matrix in our PCA. The PCA was performed with PAST version 3.16 [50].

## Results

The total area sampled within the transects was 210.4 ha (Table 3). As expected, most of the skeletal remains found in the transects belonged to terrestrial mammals larger than 5 kg in body weight. We also found bones of other vertebrates, including small mammals, dolphins, tortoises, sea turtles, and birds. Terrestrial mammalian herbivore remains dominated the DNP death assemblage. The most frequently encountered species were feral cow, feral horse, red deer, fallow deer, and wild boar (Table 4). Among the carnivores, we found bones of red fox, Egyptian mongoose, and indeterminate carnivores, but they were very rare (Table 4).

For the entire death assemblage, the NISP was 3741 and the MNI was 341 (Table 3). We estimated an average density of 17.8 specimens per hectare at DNP (NISP/ha) and an average density of 1.6 dead individuals per hectare (MNI/ha) (Table 3).

For skeletal remains from all transects, the highest NISP belongs to the red deer (NISP = 1611; 43.1%) (Table 4). In terms of the MNI, the red deer is the most commonly represented mammal (MNI = 105; 30.8%), followed by the wild boar (MNI = 91; 26.7%) (Table 4).

At DNP as a whole, most specimens were associated or articulated with other specimens from the same individual. Only 7.7% of the remains were found in complete isolation (Table 3). A high percentage of the bones were unbroken or only slightly broken (CS0 + CS1 = 66.8%). Most (99.0%) of the remains were unabraded. Unweathered (WS 0) or slightly weathered (WS 1) specimens constituted 73.8% of the sample, while those showing a weathering stage of 2 or greater represented 26.2% (Table 3). Bones displaying chewing marks constituted 23.9% of the total bone sample. Furrows were the most common type of chewing mark (63.9%) (Table 3). Marks or breakage caused by trampling were observed in 1.5% of the total sample. Ten percent of the skeletal remains were partly to extensively buried (Table 3).

Among habitats, the highest NISP occurred in the cork-oak woodland (32.3%) (Table 3). The habitat with the lowest NISP was the beach (1.7%), followed by the shrubland (2.3%) (Table 3). MNIs were highest in the Vera (18.8%) and the cork-oak woodland (17.3%) and lowest in the beach (0.9%) and active dunes (1.5%) (Table 3).

The habitat with the greatest density of remains (NISP/ha) was the cork-oak woodland (142.9), followed by the river margin (30.5) and the lake margin (29.0) (Table 3). The lowest NISP/ha was in the beach (3.6). The MNI per hectare (MNI/ha) was highest at the cork-oak woodland (7.0), followed by the lake margin (3.3), and lowest in the beach and active dunes (0.2 in both cases) (Table 3).

Associated bones dominated most of the habitats, although in the lake margin and beach, articulated bones were more abundant (Table 3). The habitats where isolated remains were moderately abundant were the beach and the meadow habitat.

Habitats displaying more complete bones (CS0 and CS1) included the marsh, the lake margin, the beach, and the dunes (Table 3). Habitats where less complete bones (CS2 and CS3) predominate were the meadow, the pine wood and the shrubland.

**Table 3. Taphonomic characterization of Doñana National Park habitats.**

| | Vera | Marshland | Lake margin | River margin | Beach | Dunes | Meadow | Pine woodland | Shrubland | Cork oak woodland | TOTAL |
|---|---|---|---|---|---|---|---|---|---|---|---|
| **Open/closed habitat** | Open | Open | Open | Open | Open | Open | Open | Closed | Closed | Closed | |
| **Sampled area (ha)** | 24.2 | 30.0 | 13.5 | 18.0 | 18.0 | 30.0 | 25.3 | 22.0 | 21.0 | 8.5 | 210.4 |
| **NISP** | 472 | 209 | 391 | 549 | 64 | 184 | 249 | 329 | 85 | 1209 | 3741 |
| **%NISP** | 12.6 | 5.6 | 10.5 | 14.7 | 1.7 | 4.9 | 6.7 | 8.8 | 2.3 | 32.3 | 100 |
| **MNI** | 64 | 19 | 44 | 32 | 3 | 5 | 50 | 45 | 20 | 59 | 341 |
| **%MNI** | 18.8 | 5.6 | 12.9 | 9.4 | 0.9 | 1.5 | 14.7 | 13.2 | 5.9 | 17.3 | 100 |
| **NISP/ha** | 19.5 | 7.0 | 29.0 | 30.5 | 3.6 | 6.1 | 9.8 | 15.0 | 4.0 | 142.9 | 17.8 |
| **MNI/ha** | 2.6 | 0.6 | 3.3 | 1.8 | 0.2 | 0.2 | 2.0 | 2.0 | 1.0 | 7.0 | 1.6 |
| **%Degree of Articulation** | | | | | | | | | | | |
| Isolated | 10.0 | 5.2 | 7.9 | 2.5 | 24.8 | 11.3 | 24.9 | 8.4 | 19.6 | 2.3 | 7.7 |
| Associated | 77.9 | 94.8 | 41.6 | 76.1 | 17.9 | 76.0 | 60.1 | 83.8 | 60.8 | 74.5 | 70.8 |
| Articulated | 12.2 | 0.0 | 50.5 | 21.4 | 57.3 | 12.7 | 15.0 | 7.8 | 19.6 | 23.2 | 21.5 |
| **%Element Completeness** | | | | | | | | | | | |
| CS 0 | 52.8 | 71.2 | 76.3 | 49.7 | 59.6 | 70.0 | 32.1 | 29.4 | 24.7 | 54.6 | 53.9 |
| CS 1 | 6.0 | 6.8 | 5.4 | 14.5 | 18.4 | 7.7 | 13.7 | 16.8 | 22.7 | 17.4 | 12.9 |
| CS 2 | 16.9 | 14.1 | 10.2 | 22.5 | 12.3 | 11.7 | 24.9 | 32.9 | 20.6 | 18.6 | 18.8 |
| CS 3 | 24.3 | 7.8 | 8.0 | 13.2 | 9.6 | 10.6 | 29.3 | 20.9 | 32.0 | 9.4 | 14.4 |
| **%Weathering** | | | | | | | | | | | |
| WS 0 | 53.1 | 70.9 | 87.2 | 40.9 | 98.3 | 90.3 | 37.4 | 43.4 | 44.7 | 29.5 | 51.2 |
| WS 1 | 14.9 | 6.9 | 6.5 | 26.0 | 0.0 | 4.5 | 32.0 | 28.0 | 43.5 | 34.8 | 22.6 |
| WS 2 | 16.0 | 10.8 | 3.6 | 15.3 | 0.0 | 1.5 | 8.7 | 13.5 | 9.4 | 19.7 | 13.1 |
| WS 3 | 4.9 | 8.9 | 2.7 | 5.4 | 1.7 | 3.3 | 19.9 | 10.3 | 1.2 | 7.8 | 6.8 |
| WS 4 | 9.8 | 2.5 | 0.0 | 12.0 | 0.0 | 0.4 | 1.9 | 4.8 | 1.2 | 7.6 | 5.9 |
| WS 5 | 1.3 | 0.0 | 0.0 | 0.4 | 0.0 | 0.0 | 0.0 | 0.0 | 0.0 | 0.6 | 0.4 |
| **%Abrasion** | | | | | | | | | | | |
| AS 1 | 100.0 | 100.0 | 100.0 | 99.8 | 83.8 | 93.7 | 100.0 | 99.7 | 100 | 99.9 | 99.0 |
| AS 2 | 0.0 | 0.0 | 0.0 | 0.2 | 14.5 | 6.3 | 0.0 | 0.3 | 0 | 0.1 | 0.9 |
| AS 3 | 0.0 | 0.0 | 0.0 | 0.0 | 1.7 | 0.0 | 0.0 | 0.0 | 0 | 0.0 | 0.1 |
| **%Trampling** | 0.2 | 1.9 | 6.4 | 0.6 | 0.0 | 0.0 | 0.4 | 0.9 | 0 | 1.6 | 1.5 |
| **%Burial** | 6.7 | 22.5 | 12.1 | 18.5 | 11.1 | 7.4 | 8.3 | 21.3 | 1.0 | 2.8 | 10.0 |
| **%Chewing marks** | 33.1 | 10.2 | 11.1 | 17.8 | 16.2 | 4.1 | 60.8 | 54.4 | 50.5 | 17.2 | 23.9 |
| **%Type of chewing marks** | | | | | | | | | | | |
| Chewed antlers | 0.5 | 0.0 | 3.0 | 0.0 | 0.0 | 0.0 | 3.4 | 1.3 | 3.8 | 1.9 | 1.7 |
| Furrows | 57.3 | 73.9 | 59.7 | 78.8 | 61.5 | 71.4 | 59.0 | 69.2 | 45.0 | 68.1 | 63.9 |
| Punctures/Pits | 10.9 | 13.0 | 20.9 | 15.9 | 30.8 | 7.1 | 13.7 | 21.4 | 8.8 | 18.5 | 16.2 |
| Scores | 27.3 | 4.3 | 13.4 | 5.3 | 7.7 | 21.4 | 21.5 | 6.0 | 35.0 | 11.5 | 15.9 |
| Breakage | 4.1 | 8.7 | 3.0 | 0.0 | 0.0 | 0.0 | 2.4 | 2.1 | 7.5 | 0.0 | 2.3 |

The percentage of each taphonomic variable refers to the NISP. Ha = hectare; CS = Completeness Stage; WS = Weathering Stage; AS = Abrasion Stage. Each of the stages is explained in the text.

Specimens displaying weathering stage 0 predominated in the marshland, lake margin, beach, and dunes. The most advanced weathering stages (WS ≥ 3) were low throughout all the DNP habitats (Table 3).

Most of the bones at DNP were unabraded, although some abraded bones occurred at the beach and the dunes (Table 3). Marks and breakage produced by trampling were rare at DNP,

**Table 4. Species composition of the death assemblages from Doñana National Park.**

| | Vera | | Marshland | | Lake margin | | River margin | | Beach | | Dunes | | Meadow | | Pine woodland | | Shrubland | | Cork oak woodland | | TOTAL | |
|---|---|---|---|---|---|---|---|---|---|---|---|---|---|---|---|---|---|---|---|---|---|---|---|
| | NISP | % | NISP | % | NISP | % | NISP | % | NISP | % | NISP | % | NISP | % | NISP | % | NISP | % | NISP | % | NISP | % |
| Feral horse | 229 | 48.5 | 0 | 0.0 | 104 | 26.6 | 2 | 0.4 | 0 | 0.0 | 1 | 0.5 | 4 | 1.6 | 1 | 0.3 | 0 | 0.0 | 35 | 2.9 | 376 | 10.1 |
| Feral cow | 37 | 7.8 | 136 | 65.1 | 21 | 5.4 | 182 | 33.2 | 1 | 1.6 | 0 | 0.0 | 0 | 0.0 | 52 | 15.8 | 2 | 2.4 | 0 | 0.0 | 431 | 11.5 |
| Red deer | 73 | 15.5 | 17 | 8.1 | 23 | 5.9 | 77 | 14.0 | 62 | 96.9 | 183 | 99.5 | 65 | 26.1 | 179 | 54.4 | 11 | 12.9 | 921 | 76.2 | 1611 | 43.1 |
| Fallow deer | 57 | 12.1 | 32 | 15.3 | 15 | 3.8 | 219 | 39.9 | 0 | 0.0 | 0 | 0.0 | 83 | 33.3 | 70 | 21.3 | 0 | 0.0 | 3 | 0.2 | 479 | 12.8 |
| Wild boar | 70 | 14.8 | 23 | 11.0 | 24 | 6.1 | 17 | 3.1 | 0 | 0.0 | 0.0 | 0.0 | 85 | 34.1 | 26 | 7.9 | 71 | 83.5 | 249 | 20.6 | 565 | 15.1 |
| Sheep | 0 | 0.0 | 0 | 0.0 | 0 | 0.0 | 24 | 4.4 | 1 | 1.6 | 0 | 0.0 | 0 | 0.0 | 0 | 0.0 | 0 | 0.0 | 0 | 0.0 | 25 | 0.7 |
| Red fox | 0 | 0.0 | 0 | 0.0 | 204 | 52.2 | 19 | 3.5 | 0 | 0.0 | 0.0 | 0.0 | 0.0 | 0.0 | 0.0 | 0.0 | 0.0 | 0.0 | 0.0 | 0.0 | 223 | 6.0 |
| Egyptian mongoose | 0 | 0.0 | 0 | 0.0 | 0 | 0.0 | 0 | 0.0 | 0.0 | 0.0 | 0.0 | 0.0 | 3.0 | 1.2 | 0.0 | 0.0 | 0.0 | 0.0 | 0.0 | 0.0 | 3 | 0.1 |
| Indeterminate cervid | 6 | 1.3 | 0 | 0.0 | 0 | 0.0 | 0 | 0.0 | 0 | 0.0 | 0 | 0.0 | 8 | 3.2 | 0 | 0.0 | 1 | 1.2 | 1 | 0.1 | 16 | 0.4 |
| Indeterminate carnivore | 0 | 0.0 | 1 | 0.5 | 0 | 0.0 | 9 | 1.6 | 0 | 0.0 | 0 | 0.0 | 1 | 0.4 | 1 | 0.3 | 0 | 0.0 | 0 | 0.0 | 12 | 0.3 |
| TOTAL | 472 | | 209 | | 391 | | 549 | | 64 | | 184 | | 249 | | 329 | | 85 | | 1209 | | 3741 | |
| | MNI | % | MNI | % | MNI | % | MNI | % | MNI | % | MNI | % | MNI | % | MNI | % | MNI | % | MNI | % | MNI | % |
| Feral horse | 16 | 25.0 | 0 | 0.0 | 12 | 27.3 | 2 | 6.3 | 0 | 0.0 | 1 | 20.0 | 2 | 4.0 | 1 | 2.2 | 0 | 0.0 | 1 | 1.7 | 35 | 10.3 |
| Feral cow | 7 | 10.9 | 3 | 15.8 | 12 | 27.3 | 9 | 28.1 | 1 | 33.3 | 0 | 0.0 | 0 | 0.0 | 2 | 4.4 | 2 | 10.0 | 0 | 0.0 | 36 | 10.6 |
| Red deer | 14 | 21.9 | 2 | 10.5 | 5 | 11.4 | 5 | 15.6 | 1 | 33.3 | 4 | 80.0 | 7 | 14.0 | 20 | 44.4 | 4 | 20.0 | 43 | 72.9 | 105 | 30.8 |
| Fallow deer | 17 | 26.6 | 4 | 21.1 | 4 | 9.1 | 8 | 25.0 | 0 | 0.0 | 0 | 0.0 | 16 | 32.0 | 8 | 17.8 | 0 | 0.0 | 1 | 1.7 | 58 | 17.0 |
| Wild boar | 7 | 10.9 | 9 | 47.4 | 9 | 20.5 | 3 | 9.4 | 0 | 0.0 | 0 | 0.0 | 23 | 46.0 | 13 | 28.9 | 13 | 65.0 | 14 | 23.7 | 91 | 26.7 |
| Sheep | 0 | 0.0 | 0 | 0.0 | 0 | 0.0 | 1 | 3.1 | 1 | 33.3 | 0 | 0.0 | 0 | 0.0 | 0 | 0.0 | 0 | 0.0 | 0 | 0.0 | 2 | 0.6 |
| Red fox | 0 | 0.0 | 0 | 0.0 | 2 | 4.5 | 2 | 6.3 | 0 | 0.0 | 0 | 0.0 | 0 | 0.0 | 0 | 0.0 | 0 | 0.0 | 0 | 0.0 | 4 | 1.2 |
| Egyptian mongoose | 0 | 0.0 | 0 | 0.0 | 0 | 0.0 | 0 | 0.0 | 0 | 0.0 | 0 | 0.0 | 1 | 2.0 | 0 | 0.0 | 0 | 0.0 | 0 | 0.0 | 1 | 0.3 |
| Indeterminate cervid | 3 | 4.7 | 0 | 0.0 | 0 | 0.0 | 0 | 0.0 | 0 | 0.0 | 0 | 0.0 | 0 | 0.0 | 0 | 0.0 | 1 | 5.0 | 0 | 0.0 | 4 | 1.2 |
| Indeterminate carnivore | 0 | 0.0 | 1 | 5.3 | 0 | 0.0 | 2 | 6.3 | 0 | 0.0 | 0 | 0.0 | 1 | 2.0 | 1 | 2.2 | 0 | 0.0 | 0 | 0.0 | 5 | 1.5 |
| TOTAL | 64 | | 19 | | 44 | | 32 | | 3 | | 5 | | 50 | | 45 | | 20 | | 59 | | 341 | |

The upper table lists the number of identified specimens (NISP). The lower table lists the minimum number of individuals (MNI).

and only at the lake margin did more than 5% of the bones show signs of trampling (Table 3). Habitats displaying the largest percentages of buried remains were the marshland, pine wood, and river margin.

In the meadow, pine woods and shrubland, bones displaying chewing marks outnumbered bones without chewing marks (Table 3), contrary to what is observed in the rest of the habitats, where non-chewed bones were more abundant. The most abundant type of chewing marks was furrows in all of the habitats (Table 3).

The results of the PCA of the proportion of taphonomic variables are shown in Fig 3. The first two components explained 89.8% of the variance (PC 1 = 76.9%; PC 2 = 12.9%). The PCA biplot suggests a separation between the active depositional environments (marshland, lake margin, river margin, beach, and dunes) and four of the non-depositional habitats (Vera, meadow, pine woodland, shrubland); the cork-oak woodland clusters with the active depositional environments but with less influence on the first two axes (Fig 3A). The active depositional habitats differ significantly from the other habitats in terms of taphonomic variables (ANOSIM test: $R = 0.57$; $p = 0.02$). The most influential variables for PC1 are bone breakage

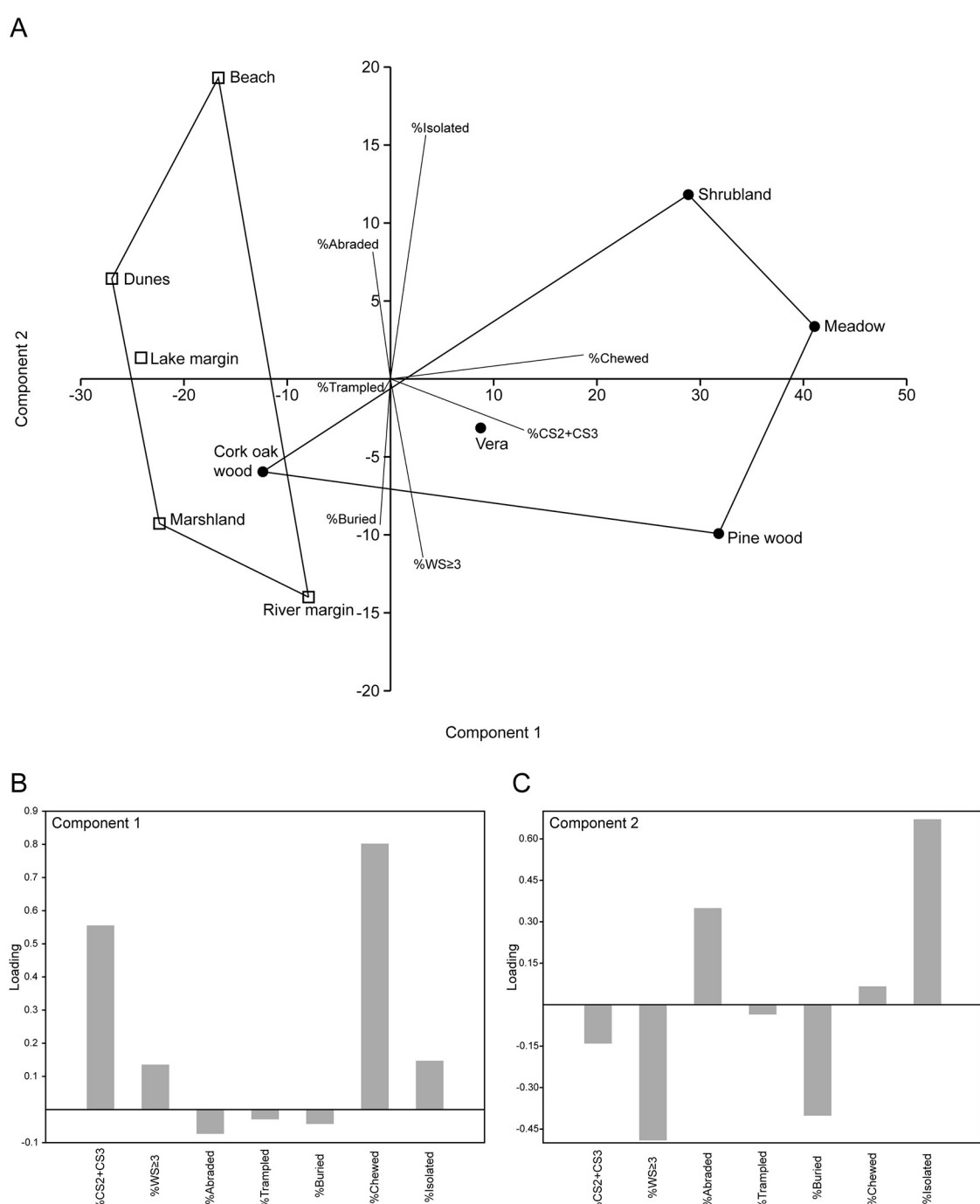

**Fig 3. Principal Component Analysis (PCA) of the ten habitats as a function of their taphonomic variables.** (A) PCA biplot showing the two first principal components (PC1 and PC2); active depositional environments represented by squares; the other habitats are indicated by circles; the correlation of each variable with each axis is indicated by the length and angle of line segments. (B) PC1 loadings; CS = completeness stage; WS = weathering stage. (C) PC2 loadings.

and the presence of chewing marks, separating at one end of the axis the habitats with more breakage (%CS2+CS3) and chewed bones from the habitats with less breakage and fewer chewed bones at the other end (Fig 3A and 3B). The most important variables for PC2 are isolated bones, at one end of the axis, and the proportion of articulated+associated bones, at the

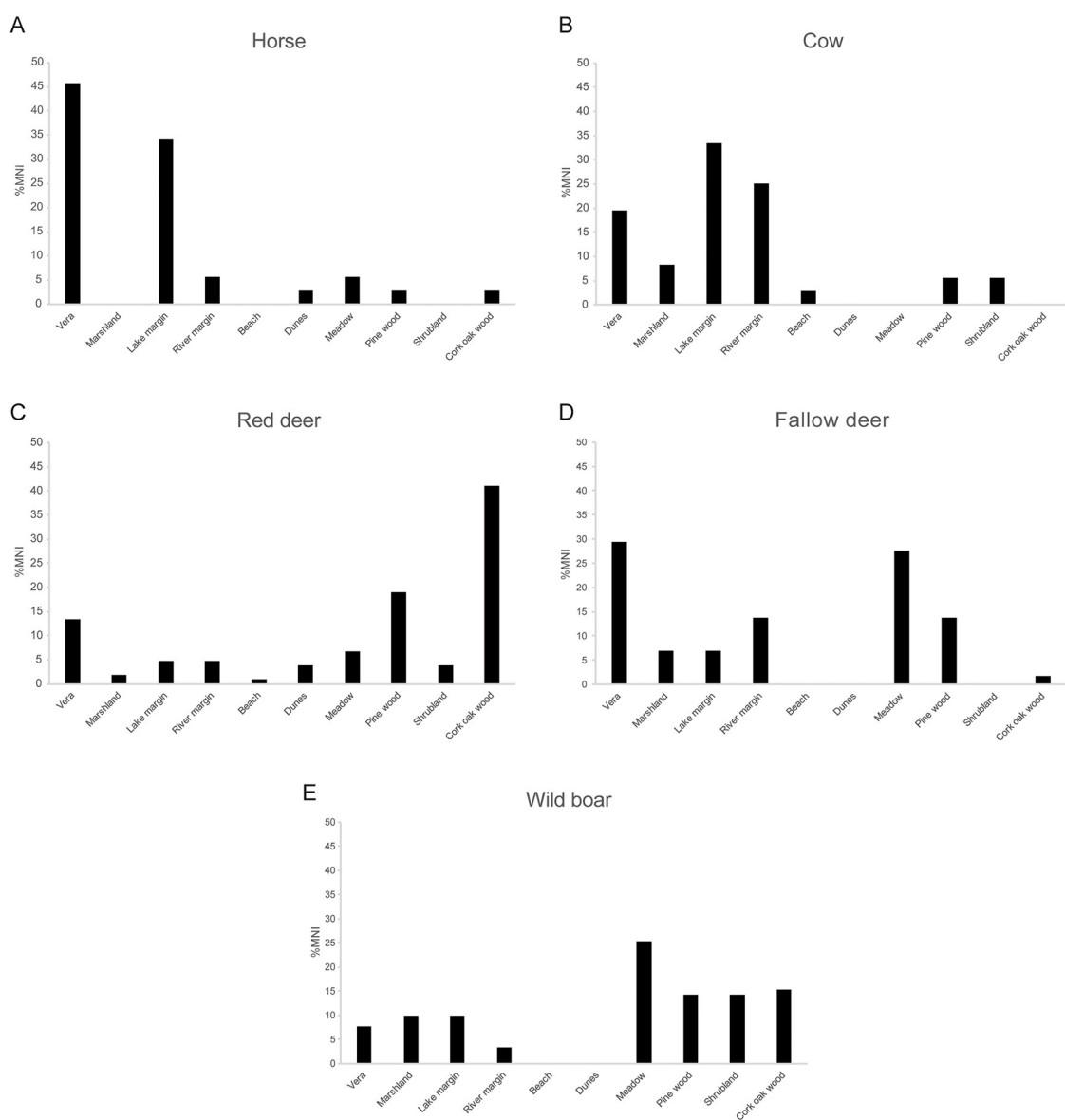

**Fig 4. Frequency by habitat of major mammals larger than 5 kg from the death assemblage of Doñana National Park.** (A) Horse. (B) Cow. (C) Red deer. (D) Fallow deer. (E) Wild boar. MNI = Minimum Number of Individuals. Raw MNI are in Table 4.

other end (Fig 3A and 3C). Other variables, including weathering stage, abrasion, and burial, also influence PC2.

In terms of the distribution of mammal remains among the different habitats (Fig 4), horses were most abundant in the Vera and lake-margin habitats. Cattle were most abundant at the lake and river margins. Red deer were, by far, most abundant in the cork-oak woodland, followed by the pine woodland. Fallow deer predominated in the Vera and meadow habitats. Wild boar were more abundant in the meadow, followed by the cork-oak woodland, shrubland, and pine woodland. One sheep was found in the beach habitat and another at the river margin (Table 4). Red fox was represented by four individuals, two found in the lake-margin habitat and two in the river-margin habitat. The only Egyptian mongoose individual present in our survey occurred in the meadow habitat (Table 4).

## Discussion

### Taphonomic characterization of DNP death assemblage: Whole-park analysis

DNP has a lower density of bones per hectare (mean NISP/ha = 17.8) (Table 3) than two documented African ecosystems. In Amboseli National Park (Kenya), the average minimum number of elements (MNE) per ha is 26.3 [9] (our estimates based on NISP are comparable with those based on the Minimum Number of Elements (MNE), given the low breakage levels at DNP). At the Virunga National Park (Democratic Republic of the Congo), Sept [18] and Tappen [19] reported an average NISP/ha of 27.4 and MNE/ha of 33.4, respectively. This outcome was expected given the lower abundance and diversity of vertebrate species over 5 kg in weight at DNP compared to the African parks.

The very large NISP/ha estimate (142.9) documented in the cork-oak woodland is striking (Table 3). This woodland is located on the northernmost border of DNP, close to the village of El Rocío. Live censuses carried out at DNP demonstrate that the density of red deer is elevated in this area compared to other areas of the park [51]. Nevertheless, it is acknowledged that the borders of protected areas experience strong human interference and may end up being populations sinks [52]. For DNP, it is known that poaching of red deer is high in this cork-oak woodland compared to inner areas of the park, which are less accessible [53], so the high NISP/ha in this habitat reflects this influence.

We obtained a MNI/ha of 1.6 for DNP as a whole (Table 3), a higher estimate than that obtained by Bernáldez Sánchez [38] at the Doñana Biological Reserve (0.3 MNI/ha). In fact, the MNI/ha that we documented would be even slightly higher (1.7) for the transects from the Doñana Biological Reserve alone (transects T1, T6, T15, T16, T19, T29, T30; Fig 1B). This difference could result from an increase in mortality on the landscape since the period when Bernáldez Sánchez [38] conducted her studies (1988–1991), but, more likely, it is the consequence of differences in approach, since Bernáldez Sánchez [38] focused on the study of more complete carcasses.

For the death assemblage as a whole, the skeletal remains were generally in a good state of preservation with little damage to the bones (i.e., a prevalence of associated, complete, and non-chewed bones) (Table 3). Since the extirpation of wolves in the 1950s, no large predators have been present in DNP. The Iberian lynx (*Lynx pardina*) is the largest mammalian predator now present in this area. Although this lynx has been reported to kill and feed on red-deer fawns, fallow-deer fawns, and young wild boar, its most common prey is the rabbit, which constitutes more than 70% of its diet [54,55]. The DNP Iberian lynx population has significantly declined since the 1970s–80s (when the ungulate killings were reported), following a significant decrease of rabbits, so its predatory effect on wild ungulate populations must have greatly diminished in the past 30 years [56]. The inferred very low predator-to-prey ratio at DNP results in bones surviving longer than in ecosystems with a greater number of species and individuals of native predators [9,18,19]. We recognize that the low incidence of natural predation on ungulate species at DNP is not typical of natural systems. However, this circumstance might allow us to evaluate the taphonomic impacts of other causes of mortality, such as disease or weather events.

### Taphonomic characterization of DNP death assemblage: By-habitat analysis

When we analyzed habitats independently, we observed that three habitats had higher levels of bone breakage (CS2+CS3). These habitats are the meadow, the pine woodland, and the

**Table 5. Correlation tests between different taphonomic variables from the different habitats in Doñana National Park.**

|  | %CS2+CS3 | %WS≥3 | %Abraded | %Trampled | %Buried | %Chewed | %Isolated |
|---|---|---|---|---|---|---|---|
| %CS2+CS3 |  | 0.08 | 0.58 | 0.48 | 0.51 | *0.001* | 0.40 |
| %WS≥3 | 0.58 |  | 0.52 | 0.44 | 0.84 | 0.17 | 0.45 |
| %Abraded | -0.20 | -0.23 |  | 0.29 | 0.68 | 0.48 | 0.90 |
| %Trampled | -0.25 | 0.28 | -0.37 |  | 0.11 | 0.70 | *0.02* |
| %Buried | -0.24 | 0.07 | 0.15 | 0.53 |  | 0.60 | 0.43 |
| %Chewed | 0.89 | 0.47 | -0.25 | -0.14 | -0.19 |  | 0.37 |
| %Isolated | 0.30 | -0.27 | 0.05 | -0.72 | -0.28 | 0.32 |  |

p-values are in the upper right above the diagonal, significant correlations are marked in italics (α = 0.05); Spearman's $r_s$ are in the lower left below the diagonal.
CS = Completeness Stage; WS = Weathering Stage.

shrubland (Table 3). Breakage during the biostratinomic phase might be related to chewing, trampling, transport or advanced levels of weathering. A correlation analysis among different taphonomic variables showed that breakage (CS2+CS3) had a significant positive correlation with the incidence of chewing marks (Table 5). The PCA, which separated habitats as a function of the degree of breakage and the incidence of chewing marks on PC1, showed the same pattern: the meadow, pine woodland and shrubland display the highest incidence of breakage and chewing marks (Fig 3).

Since predation pressure at DNP is low, chewing marks are most likely caused by scavengers. The primary mammalian scavengers are the red fox (*Vulpes vulpes*), the Egyptian mongoose (*Herpestes ichneumon*), and the wild boar (*Sus scrofa*). Among birds, three species of vulture are present at DNP: the griffon (*Gyps fulvus*), the Egyptian vulture (*Neophron percnopterus*), and the black vulture (*Aegypius monachus*). None of these vulture species have breeding populations in the park, although they are regular visitors year round. It was beyond the scope of this study to perform a detailed analysis of chewing marks in order to identify which scavengers produced them, but the most common type of marks are conspicuous (furrows, followed by punctures/pits and scores; Fig 5). In their actualistic analysis of damage produced by vultures, Domínguez-Solera and Domínguez-Rodrigo [57] reported that, although vultures are able to produce punctures and pits similar to those produced by mammalian carnivores, the most abundant damage patterns that they produce are inconspicuous, requiring magnification to properly identify them. These authors also observed that many bones were unbroken, whereas in our study, there was a significant correlation between the frequency of chewing marks and the degree of fragmentation (Table 5). Behrensmeyer and Dechant Boaz [9] reported that vultures in Amboseli National Park usually left bones of large vertebrates intact and mostly articulated. During the sampling of one of our Vera transects, we observed many vultures flying and landing close to our location. After the vultures were gone, we approached the area where they had congregated. They had been feeding on a red-deer carcass. Although they had disarticulated some skeletal remains, the majority of the bones were articulated and barely damaged. In addition, chewing marks are abundant in bones from the pine woods and the shrubland, two closed habitats, whereas vultures preferentially scavenge in open areas where the visibility of carcasses is good.

Vicente et al. [58] provided evidence that the red fox and, especially, the wild boar are the most common scavengers in DNP. These authors positioned camera-traps on 10 red-deer carcasses to monitor the activities of different scavengers with the aim of evaluating the spread of tuberculosis among the DNP fauna. Wild boars were always present around the carcasses and scavenged them in 80% of their observations. The red fox was present at 80% of the carcasses

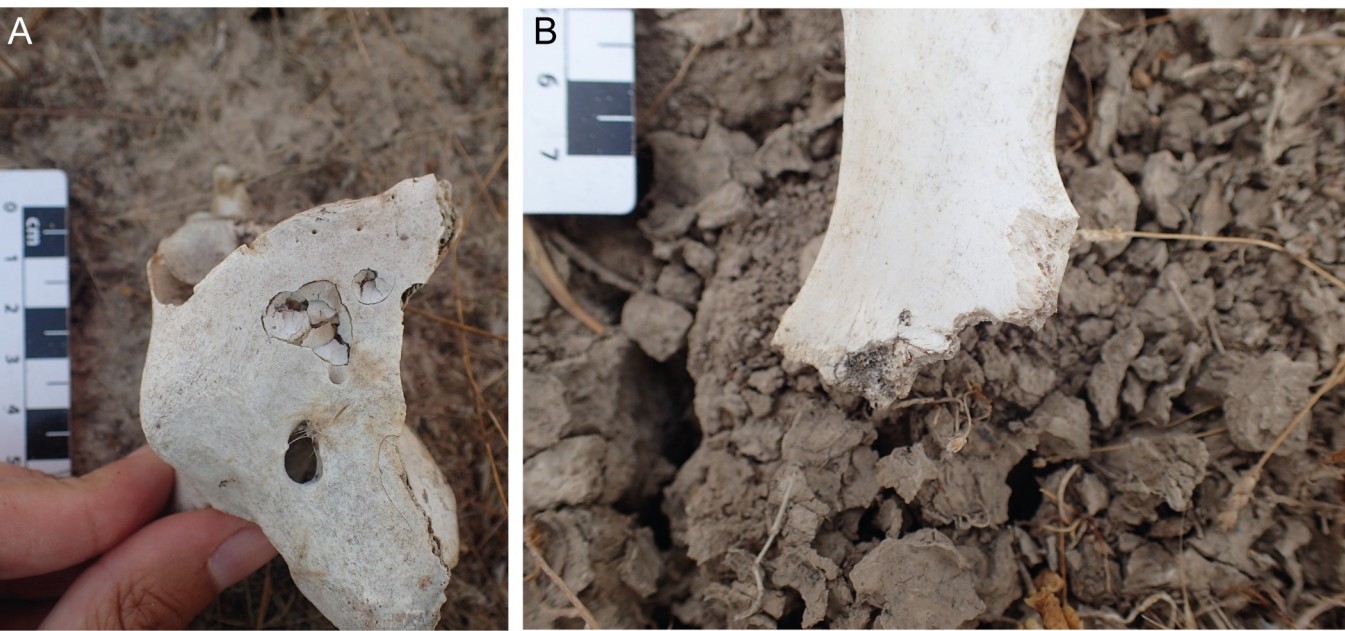

**Fig 5. Chewing marks on DNP bones.** (A) Red-deer atlas showing large punctures. (B) Close-up view of a wild-boar ischium showing furrows and a small puncture.

and scavenged them in 30% of the cases. The Egyptian mongoose scavenged only one of the carcasses. Actualistic taphonomic studies of bone modifications produced by suids reveal that these animals are capable of producing conspicuous chewing marks, very similar to those documented for dogs and hyaenas [59–61]. Studies of the damage produced by red fox indicate that the size of pits, punctures and scores is usually <2.5 mm in width [62–64]. Based on these observations, we attribute large punctures, such as those observed in Fig 5A, to wild boars, whereas small punctures, such as the one observed in Fig 5B, could be from the red fox. In summary, we consider that wild boar and red fox are the main scavengers, with the former being the prevalent scavenging agent at DNP.

Bone assemblages in active depositional environments differ statistically from those in other habitats, although their difference is caused not by the expected variable, the percentage of buried bones, but by the frequency of breakage and chewing (Fig 3A and 3B). Active depositional environments show a lower incidence of breakage and chewing. Without information from the live censuses, it is difficult to determine whether the reason for this separation of habitats is caused primarily by the presence or absence of scavengers among the different habitats. Habitats such as the beach and dunes are likely non-preferred habitats for all of the taxa sampled in our bone survey (including the scavengers), as these habitats do not offer resources such as water, food, or shade. Thus, the low frequency of scavengers may be the reason for the low levels of breakage and chewing in these two habitats. However, other factors are more likely in other habitats. For example, the low levels of breakage and chewing marks observed in the cork-oak woodland could be the consequence of partial consumption of skeletal remains by scavengers, given the high abundance of carcasses in this habitat (Table 3). The same might be true in the lake margin or the river margin. The marshland is seasonally covered by shallow water, which might favor low levels of bone alteration, including the action of scavengers.

The beach, meadow and shrubland show a higher abundance of isolated bones (Fig 3, Table 3). At the meadow and shrubland, the isolation (scattering) of skeletal remains could be

the consequence of scavengers. At the beach, wave action could have played a role in disarticulating and dispersing the bones. The beach had the highest degree of abrasion of bones. Abrasion was also notable on bones from the dunes. This was expected as bones in these habitats are subject to interactions with sedimentary particles through waves and wind.

The highest abundance of bones in the process of being buried occurred at the marshland and river margin, two active depositional environments, and interestingly, in the pine woodland. Burial in the pine woodland occurs mainly by the growth of vegetation over the bones. We expected higher levels of burial at the beach and dune habitats than those we observed (Table 3). It could be that complete burial occurs in these habitats faster than in other habitats and therefore went undetected.

Weathering Stages 0 and 1 predominated (>65%) in all habitats. Following Behrensmeyer [46], this finding suggests that most of the remains currently present at the DNP surface come from animals that died in the last 2.5 years. It could also be the case that these weathering stages occur at different rates in DNP compared to Amboseli National Park, where the weathering scale was developed [46], given their different climatic regimes. Weathering stages of 3 or more (WS≥3) reach greater frequency (15%–25%) at the Vera, river margin, meadow, pine woodland and cork-oak woodland, representing the oldest surface remains at DNP.

Trampled bones were rare at DNP, with the greatest incidence in the lake margins (Table 3). This process is evident from the parallel, fine scratches on the bone surfaces, the presence of adjacent bone fragments, and the high angles that bones assume in the burial process (Fig 6). The same features were observed by Behrensmeyer and Dechant Boaz [9] in the swamp and lake-bed habitats at Amboseli National Park and by Haynes [16] in his study of bone accumulations around water holes located in northern North America and southern Africa. During the summer months at DNP, water becomes scarce, so animals concentrate around the shrinking water bodies, such as these lakes, and walk over the skeletal remains present in these areas. As in the study of Behrensmeyer and Dechant Boaz [9], we consider that a major process of burial of bones at the DNP lake margin/bed habitat is trampling. The negative correlation between the frequency of trampling and the frequency of isolated bones (Table 5) could indicate that the more isolated and scattered the bones are, the less likely it is that an ungulate encounters and tramples on them.

## Distribution of species in the live and death assemblages

Although a quantitative comparison of the fidelity of the death assemblage to the live assemblage was beyond the scope of this study, it was evident that the most common herbivorous mammals documented are those that are also more abundant in the living populations at DNP [40]. For birds, it was striking how few bones we found despite the presence of species of considerable body size at DNP (Table 2). In her study of avian bone density, Dumont [65] found that bird skeletons are stronger and stiffer than the skeletons of similar-sized mammals. Nevertheless, this stiffness corresponds to more brittle bone behavior, which is more prone to fracturing than the mammalian bones [65]. Cruz [23] and Behrensmeyer et al. [66] found that bird bones undergo faster deterioration by weathering than mammal bones. The higher tendency to breakage and disintegration due to weathering might be the causes behind the rarity of bird bones in the DNP death assemblage.

For the time being, we can qualitatively compare our observations with what is known about the distribution, habitat and resource use of the ungulates at DNP. The comparison is hindered by the fact that most studies of living ungulates have been conducted within the Doñana Biological Reserve instead of over the entire National Park, so some habitats (e.g., river margin, beach, meadow) are not well documented in terms of occupancy and resource

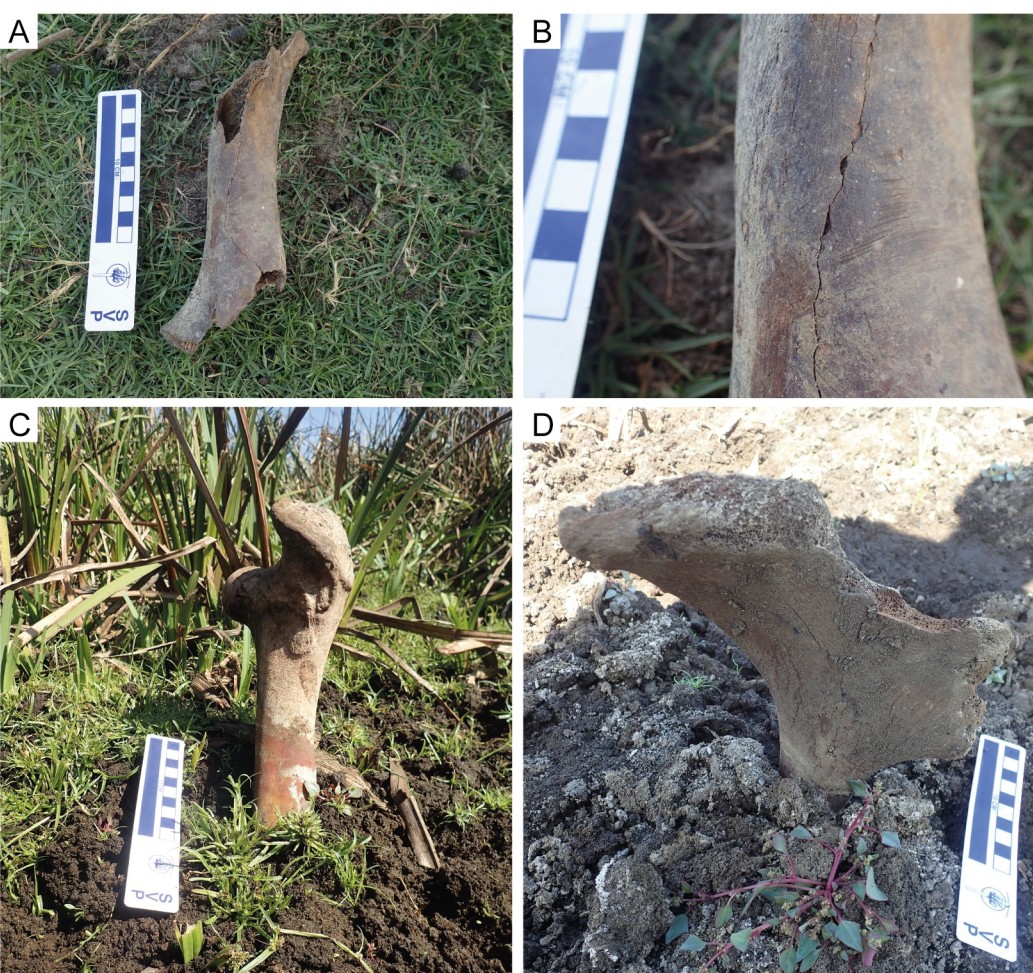

**Fig 6. Trampled bones in the lake-margin habitat.** (A) Cow or horse humerus with trampling marks. (B) Close-up view of (A) where the fine, parallel scratches produced by trampling are visible. (C) Cow femur half buried and displaying a near-vertical orientation. (D) Cow innominate half buried and displaying near-vertical orientation.

use by resident mammals. Our proposed long-term bone survey will have the potential to contribute information about the mammalian population dynamics in these less-studied habitats.

The three wild ungulates from DNP are the fallow deer, red deer and wild boar. The fallow deer is mainly a grazer and lives in close association with pastures of the Vera and meadows [67–69]. This habitat preference in life is reflected in the death assemblage (Fig 4, Table 4). The fallow deer occurs in the death assemblage with moderate abundance in one unexpected habitat: the pine woodland (Fig 4). This unexpected abundance might be the consequence of the proximity of one of the three pine-woodland transects (T13) to meadow habitats where pasture is abundant and where numerous live fallow deer individuals were observed (Fig 1B). The fallow deer might use this contiguous pine woodland as a resting or hiding area rather than a feeding area.

The red deer has a more varied diet than the fallow deer, with more browsing tendencies [68]. This versatility is reflected in the death assemblage, with the presence of red deer in most habitats from DNP, including the active dunes (Fig 4). Its browsing behavior may also be reflected in the death assemblage, as the highest MNI for red deer occur in the cork-oak and pine woodlands (Fig 4, Table 4).

The wild boar has an omnivorous diet that includes fungi, rhizomes and other plant foods, vertebrates, and invertebrates, and is cosmopolitan in terms of the habitats where it feeds [67,70]. Venero Gonzales [67] reported that wild boar feeds in the marshes, the pine woods, the Vera pastures, the shrubland, and the pastures close to lake margins. In the death assemblage, we found a balanced distribution of the wild-boar remains among habitats. Although it is slightly more abundant in the meadow, pine woods, shrubland, and cork-oak woodland, it is also present in the Vera, marshland, and lake margin (Fig 4).

Among feral animals, cattle in Doñana mainly feed on green grasses and rushes, so their preferred habitats are the Vera, and the pastures near the lakes and river [71]. We tracked the movements of five cows with GPS radio-collars (unpublished data), and can confirm that cows mainly occupied the habitats previously mentioned. The cow death assemblage reflected the expected habitat occupancy (Fig 4). The feral horses are grazers that tend to stay in the marshes, Vera, and lake margins [71]. The horse death assemblage showed the expected frequencies for the Vera and lake margin; however, the absence of horse skeletal remains in the marshes differs from the abundance of horses in the live assemblage (Fig 4, Table 4). Cow and horse are the most common species in the lake-margin habitat (Table 4). In fact, during our lake-margin transects, live cattle and horses were commonly observed. These species are heavy grazers that have high demands for drinking water and linger around water bodies. Their congregation around the lakes is the likely cause of the trampling of bones from this habitat.

Although their numbers have greatly declined in recent years in DNP, live sheep mainly occupy the marshes [71]. In the death assemblage, they are present in very low numbers and only at the river margin and beach, locations unexpected for this species.

The low numbers of NISP, MNI and density (NISP/ha, MNI/ha) of dead animals found in the beach and dunes seem to reflect the infrequent use of these areas by the live animals. These habitats do not offer water, food, or good resting places, at least, for the animals in this study. In the future, a quantitative live-dead fidelity study will shed more light on the degree of concordance between the living fauna and the death assemblages.

## Ecological information in the study of modern death assemblages

Several studies have emphasized the potential of modern death-assemblage monitoring to provide useful ecological information about the species under study [e.g., 4, 5]. Miller [13] used the distribution of shed antlers of *Cervus elaphus* in Yellowstone National Park (Wyoming, USA) to study the habitat utilization of this species. Similarly, the distribution of the shed antlers at DNP provides insights about cervid habitat use. The DNP meadow habitat (represented by three transects, T12, T18, and T22 in the area known as Las Marismillas to the south of DNP; Fig 1B) has the highest concentration of shed antlers (47% of all antlers found at DNP; in the other habitats the frequency never exceeds 15%). Twelve antlers belonged to red deer, three to fallow deer, and 18 were not identifiable to either species because they were from young individuals. For these deer, only the males possess antlers, and in both species the antlers are shed at the end of the winter [69,72]. Miller [13] stated that, given the low probability of long-distance transport of these heavy skeletal elements, their abundance in specific places reveals preferred areas for male deer to reside during the late winter. With the same reasoning, these meadows at DNP seem to be a preferred habitat during the late winter for male cervids. We are not aware of any study documenting locations or habitats where male cervids congregate in the late winter at DNP. This finding is one indication that the DNP death assemblage has the ability to provide ecological information.

## Doñana National Park death assemblage and the fossil record

At DNP, where varied habitats and depositional processes are present in a relatively small area, we can evaluate the potential of each of its habitats to preserve skeletal remains and, over time, to produce vertebrate fossil sites. In taphonomy, no single conjunction of factors guarantees fossilization and, as mentioned before, here we are evaluating the biostratinomic phase; diagenesis (the post-burial phase) ultimately determines whether fossilization occurs or whether skeletal remains are recycled and lost to the rock record. In any event, for preservation, three factors seem to be important in the biostratinomic phase: the abundance of remains, the completeness (integrity) of remains, and their burial. A fair to good abundance and integrity of remains are needed [73]. If remains are rare and their integrity is deficient, they will have reduced chances to survive to burial and diagenetic processes. Since burial is an obligatory step for fossilization, only habitats where burial occur will contribute to the fossil record [73]. Otherwise bones remain exposed until they disintegrate.

We evaluated the degree to which the habitats at DNP fulfill the three requirements. Fig 7 shows the following taphonomic variables for each of the habitats: NISP/ha to measure the abundance of skeletal remains, %CS0+CS1 to measure the completeness of the remains and % Buried remains to account for the burial process. For these three taphonomic variables, we used the average measures for the park as a whole as our threshold values (i.e., NISP/ha = 17.8, %CS0+CS1 = 67% and %Buried = 10%; Fig 7). We consider that the habitats with the highest fossilization potential at DNP are those for which all three variables are above those average estimates. The only habitat that fulfills the requirements of having abundant remains with high element completeness and undergoing burial, as shown by estimates that surpass the DNP average values, is the lake margin (Fig 7). The river margin is close to meeting the three requirements. The abundance of remains and frequency of burial are above the average threshold value. The only requirement not met by the river-margin remains is the degree of completeness, although it is quite close to the DNP threshold value (%CS0+CS1 = 64% for the river margin *vs* %CS0+CS1 = 67% for the average DNP). It is worth highlighting that many

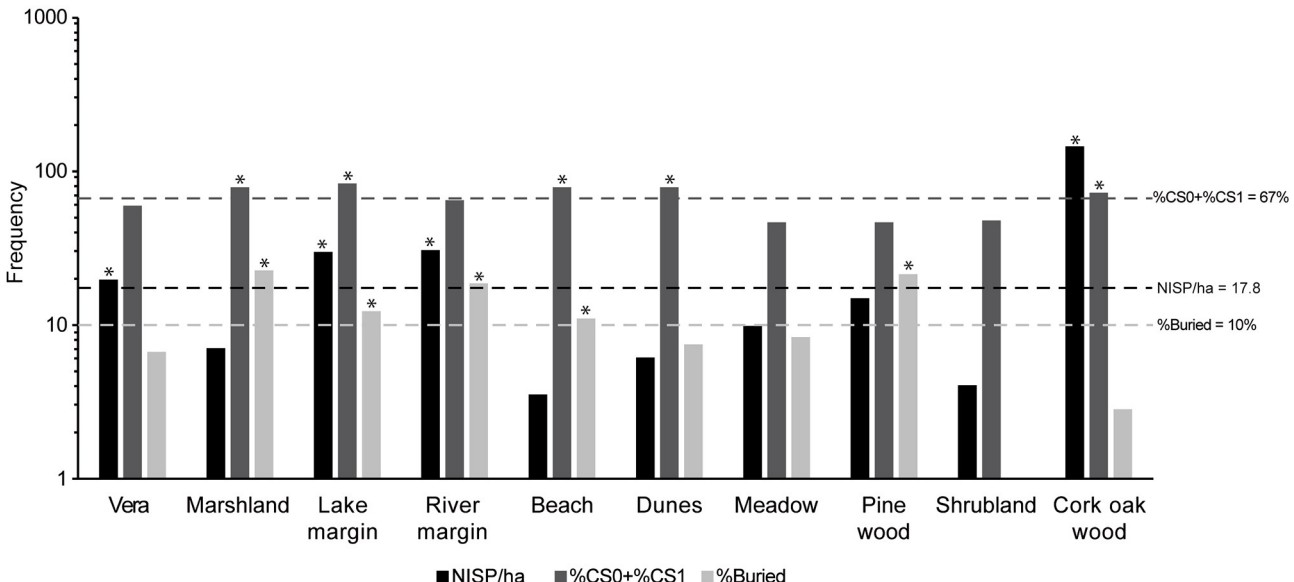

**Fig 7. Biostratinomic taphonomic variables that favor the formation of a fossiliferous site.** NISP/ha = Number of Identified Specimens per hectare; CS = Completeness stage (see text for the meaning of each stage). Dashed lines mark the average values for DNP. Asterisks mark the instances in which habitat values exceed the DNP average values.

vertebrate fossil sites occur in sedimentary facies corresponding to lake margin and river-margin habitats [e.g., 74]. In due time, if diagenesis does not eliminate the skeletal remains, these attritional death assemblages could give rise to time-averaged fossil sites.

Other habitats fulfill at least two requirements but are far from the remaining one. For the marshland and beach, skeletal remains are complete or almost complete and the burial process is occurring (Fig 7). Nevertheless, remains are rare, so they would produce at best poor fossil localities. For the cork-oak woodland, although remains are abundant and well preserved, they are being buried at a low rate. Therefore, bones will remain exposed for a longer time and their destruction/recycling before burial is likely.

As indicated before, the anomalous property of a near absence of predators differentiates our bone survey from previous surveys. The DNP completeness and bone-damage values are free from the predator imprint making them useful for assessing bone modifications from other agents on fossil assemblages. Another interesting finding, in this case related to scavenger action, is that active depositional environments showed lower levels of scavenging compared to the other habitats (Fig 3, Table 3). We could then consider the scavenging levels from active depositional environments (i.e., those that are prone to be preserved in the fossil record) as a minimum baseline value. In turn, habitats that are not prone to fossilization might have higher values of scavenging, but these will not be reflected in the fossil record.

Modern bone surveys have the potential to provide useful information to understand the past (formation of deep-time to young fossil records), the present (modern ecological information) and, even, the future (expected fossil assemblages). In this last regard, Plotnick and Koy [75] addressed the question of Behrensmeyer et al. [76], "What will be the future fossils of our age, and how will they be preserved?" They concluded that the "Anthropocene" biostratigraphic unit in the terrestrial realms will mainly consist of fossils of a cosmopolitan fauna of humans and their domestic animals (favored by the burial of humans in cemeteries and of domestic species in landfills and pits). These authors indicate that the chance for wild animals to become part of the fossil record will be very low. We add to this last statement that the sedimentary facies representing active depositional environments from protected natural areas (such as those of Doñana National Park) and from remote areas on Earth will constitute the last strongholds of wild animal remains that future vertebrate paleontologists will be able to study.

## Conclusions

We synthesized the results of the actualistic taphonomic study of modern carcasses and bones from Doñana National Park, located in a coastal Mediterranean ecosystem in Andalusia, Spain. Although many fossil sites have formed under a Mediterranean climatic regime, this is one of the few taphonomic baseline studies for this type of environment.

We conducted bone surveys in ten different habitats from Doñana National Park, half of them occurring in active depositional environments. Most of the recorded bones belonged to land mammals larger than 5 kg, which at Doñana mainly correspond to ungulates. The best-represented species in terms of the total number of bones and individuals found is the red deer.

No large predators have been present in Doñana National Park since the 1950s, and the low predation pressure results in generally good preservation of bones compared to other natural areas. The near absence of vertebrate predators at DNP makes our bone survey useful to better understand the effect and taphonomic imprint of other biostratinomic agents. For example, the significant correlation between the incidence of bone breakage and the incidence of

chewing marks implies that scavenging is the major agent of bone breakage. The wild boar is the most important scavenger at Doñana National Park.

We evaluated similarities and differences of taphonomic variables among habitats and found that active depositional environments differ from the other habitats. Although, as expected, the process of burial was prevalent in the active depositional environments, their differentiation from other habitats was mainly caused by the degree of breakage and the presence of chewing marks, which are higher in the non-depositional environments. We determined that the lake margin and the river margin show the highest potential to become fossil sites at the biostratinomic level: their death assemblages contain abundant remains which display fair to good completeness and are undergoing burial. These locations correspond to the sedimentary facies in which much of the vertebrate fossil record is found.

A qualitative comparison between the faunal composition and abundance of the live and dead assemblages indicates good concordance: the highest NISP and MNI were observed in habitats preferred by many of the ungulates. In these habitats, feeding opportunities, water supplies, and resting areas are abundant. Skeletal remains are rare in habitats where these animals spend little time, such as the beach and the active dunes.

Taphonomic actualistic studies can provide useful ecological information, such as habitat use by different species. Based on the concentration of cervid shed antlers, we determined that the meadow habitat is a preferred location for male deer in the late winter.

Taphonomic studies of vertebrate skeletal assemblages in modern landscapes cannot fully predict the likelihood of fossilization as remains disappear from our sight during diagenesis. Nonetheless, these studies provide invaluable information about the myriad of biostratinomic factors at work and can help us disentangle varied biotic and abiotic agents of modification. The more actualistic taphonomic studies that we conduct and the more varied the regions and climatic regimes that we survey, the more insightful taphonomic histories we will be able to reconstruct for past, present or even future faunal records.

## Acknowledgments

MSD thanks David Paz (EBD-CSIC) and Miguel Ángel Bravo (EBD-CSIC) for their invaluable help during our first visit to Doñana National Park and continuous support. MSD also thanks Anna K. Behrensmeyer for the invitation to participate in the Amboseli 2016 bone survey; field methods used in this paper were learned there. Dr. Eloísa Bernáldez Sánchez provided advice about the Doñana death-assemblage study. Isabel Afán (LAST-EBD service) put together the map shown in Fig 1B. DMMP is a predoctoral fellow of the 'Formación Personal Investigador' program associated to Project CGL2015-68333-P. This study is part of the UCM 910607 research group.

## Author Contributions

**Conceptualization:** M. Soledad Domingo, Catherine Badgley.

**Data curation:** M. Soledad Domingo, David M. Martín-Perea, Catherine Badgley, Enrique Cantero, Paloma López-Guerrero, Adriana Oliver, Juan José Negro.

**Formal analysis:** M. Soledad Domingo.

**Funding acquisition:** M. Soledad Domingo, Catherine Badgley, Juan José Negro.

**Investigation:** M. Soledad Domingo, David M. Martín-Perea, Catherine Badgley, Enrique Cantero, Paloma López-Guerrero, Adriana Oliver, Juan José Negro.

**Methodology:** M. Soledad Domingo, Catherine Badgley.

**Project administration:** M. Soledad Domingo, Juan José Negro.

**Writing – original draft:** M. Soledad Domingo.

**Writing – review & editing:** M. Soledad Domingo, David M. Martín-Perea, Catherine Badgley, Enrique Cantero, Paloma López-Guerrero, Adriana Oliver, Juan José Negro.

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
