## [Decision Letter · Decision Letter 0]

30 Sep 2020

PONE-D-20-23069

Taphonomic information from the modern vertebrate death assemblage of Doñana National Park, Spain

PLOS ONE

Dear Dr. Soledad Domingo,

Thank you for submitting your manuscript to PLOS ONE. After careful consideration, we feel that it has merit but does not fully meet PLOS ONE’s publication criteria as it currently stands. Therefore, we invite you to submit a revised version of the manuscript that addresses the points raised during the review process.

Please find below comments and suggestions by two reviewers. Both suggested minor revisions. Please check the few suggestions and re-submit a revised version of your interesting maniscript. The manuscript is detailed and of broad interest. I agree with the reviewers' opinions. Thank you for your submission.

We look forward to receiving your revised manuscript.

Kind regards,

Luca Pandolfi

Academic Editor

PLOS ONE

Additional Editor Comments:

Dear Authors,

please find attached comments and suggestions by two reviewers. Both suggested minor revisions. Please check the few suggestions and re-submit a revised version of your interesting maniscript.

Thank you very much for your submission.

Sincerely

Journal Requirements:

2. We note that Figure 1 in your submission contain map/satellite images which may be copyrighted. All PLOS content is published under the Creative Commons Attribution License (CC BY 4.0), which means that the manuscript, images, and Supporting Information files will be freely available online, and any third party is permitted to access, download, copy, distribute, and use these materials in any way, even commercially, with proper attribution. For these reasons, we cannot publish previously copyrighted maps or satellite images created using proprietary data, such as Google software (Google Maps, Street View, and Earth). For more information, see our copyright guidelines: http://journals.plos.org/plosone/s/licenses-and-copyright.

2.1.    You may seek permission from the original copyright holder of Figure 1 to publish the content specifically under the CC BY 4.0 license. 

2.2.    If you are unable to obtain permission from the original copyright holder to publish these figures under the CC BY 4.0 license or if the copyright holder’s requirements are incompatible with the CC BY 4.0 license, please either i) remove the figure or ii) supply a replacement figure that complies with the CC BY 4.0 license. Please check copyright information on all replacement figures and update the figure caption with source information. If applicable, please specify in the figure caption text when a figure is similar but not identical to the original image and is therefore for illustrative purposes only.

Reviewers' comments:

Reviewer's Responses to Questions

**Comments to the Author**

1. Is the manuscript technically sound, and do the data support the conclusions?

Reviewer #1: Yes

Reviewer #2: Yes

2. Has the statistical analysis been performed appropriately and rigorously? 

Reviewer #1: Yes

Reviewer #2: Yes

3. Have the authors made all data underlying the findings in their manuscript fully available?

Reviewer #1: No

Reviewer #2: Yes

4. Is the manuscript presented in an intelligible fashion and written in standard English?

Reviewer #1: Yes

Reviewer #2: Yes

5. Review Comments to the Author

Reviewer #1: Congratulations on an outstanding contribution. This is excellent research, extremely valuable, and well done. I find no fault with the methods or text, and the references are appropriate. I list below a few typos that I picked up. Other than that, the data are clearly presented, and the appropriate statistical analyses applied. The results are new and very interesting. I find the paper brilliant and of interest to a broad scientific community.

A previous dropdown bar: 3. Have the authors made all data underlying the findings in their manuscript fully available? I replied no. I did not see any supporting information.

Fig. 7. Frecuency change to frequency

Line 59. Weigelt conducted one of the first studies….

Table 1. This habitat is 30 km long….. shrubland where carried out…

Line 152. Taphonomic variables

Line 153. We consider therefore that their study and our studies are not redundant but complementary.

Line 172. Projected the 1 km length…

Table 2. How many constitutes abundant, 10, 50?

Line 275. Furrows (scalloped edges) were…… Furrows and scalloped edges are different taphonomic features, so eliminate the latter or add it to your definition of a furrow.

Line 304. Rest of THE habitats

Line 425. The majority of the bones ….

Line 511. Nevertheless, …

Reviewer #2: This study characterizes predominantly modern mammal bones and carcasses in Donana National Park in Spain as an example of extant actualistic taphonomic analyses. Overall the study is very well organized and written.

I have a few notes:

Line 82: In reference to Tzedakis 2007, it would be helpful to expand on the differences between the Tzedakis study and the other studies in regard to the age of the onset of the Mediterranean climate regime as it is directly pertinent to this study. This would be especially helpful in regards to understanding the different criteria upon which these inferences were made.

Methods: Did you collect data on bone orientation? This would be especially useful and important for shoreline environments. Even in other environments, describing the orientations of bone long axes provides a context and process for disarticulation in different environments, especially during decomposition and other early-stage biostratinomic processes.

The authors mention that they utilized Behrensmeyer's weathering scale based on Ambolseli work. As they correctly note, this scale is specific to the tropical savannah conditions of Kenya. While the authors mention this caveat, I encourage them to discuss more about the potential differences and modifications to this scale that may be a better tool for a Mediterranean study.

6. PLOS authors have the option to publish the peer review history of their article (what does this mean?). If published, this will include your full peer review and any attached files.

Reviewer #1: **Yes: **Lucinda Backwell

Reviewer #2: **Yes: **Joseph E. Peterson

---

## [Author Response · Author response to Decision Letter 0]

15 Oct 2020

PONE-D-20-23069

Taphonomic information from the modern vertebrate death assemblage of Doñana National Park, Spain

PLOS ONE

Dear editor,

Thank you for all your comments. We have corrected the map issue that you pointed out. We replaced the problematic map (Figure 1B) with a map that complies with the CC-BY 4.0 licence. We have indicated the source of the map and this compliance in the figure caption. The document that certifies the compliance is included in our submission under the "Other" tab. 

Also, please find below our responses to the reviewer's suggestions and concerns. We made many of the changes/corrections suggested. Otherwise, we provided an explanation for why we prefer to leave the text as it is. 

Finally, we are aware that Plos One does not recommend including editors or reviewers in the acknowledgment section so we would like to thank you here for your editorial work and the reviewers, Lucinda Backwell and Joseph E. Peterson, for their helpful comments. 

Sincerely,

M. Soledad Domingo (on behalf of all authors)

Universidad Complutense de Madrid

REVIEWER #1

Reviewer #1 answered "No" to question 3 about making data from the article fully available based on the absence of supporting information. No supporting information was included in this article because we provide all the data underlying the results in the paper in Tables 3 and 4. The raw numbers for each analyzed habitat are provided in these tables (i.e., NISP, MNI, hectares). So, all of the results and proportions can be readily obtained within the manuscript, and we consider that supporting information is not needed.

Other concerns or suggestions of reviewer #1 are listed below in italics and our response is in regular font:

• Fig. 7. Frecuency change to frequency We corrected the typo.

• Line 59. Weigelt conducted one of the first studies…. We do not understand this correction as our text coincides with what the reviewer suggests. We left the sentence as it was.

• Table 1. This habitat is 30 km long….. shrubland where carried out… We corrected the typos.

• Line 152. Taphonomic variables We corrected this.

• Line 153. We consider therefore that their study and our studies are not redundant but complementary. We followed the reviewer's suggestion.

• Line 172. Projected the 1 km length… We corrected this.

• Table 2. How many constitutes abundant, 10, 50? We obtained data for mammals from references 40 and 41 and for birds from references 42, 43 and 44, as indicated in the table caption. These authors do not report specific numbers about what value is considered to be 'abundant' so we cannot provide a number either. In any event, it may be different for each of the species reported due to their different population densities so it might not even be possible to provide a unique number for all of the species listed. We leave the table as it is so that the readership has a qualitative (not a quantitative) idea of species abundances at Doñana National Park. 

• Line 275. Furrows (scalloped edges) were…… Furrows and scalloped edges are different taphonomic features, so eliminate the latter or add it to your definition of a furrow. We eliminated '(scalloped edges)' as suggested by the reviewer.

• Line 304. Rest of THE habitats We corrected this.

• Line 425. The majority of the bones …. We corrected this.

• Line 511. Nevertheless, … We corrected the typo.

REVIEWER #2

Reviewer's #2 comments are indicated below in italics and our reponses in regular font:

Line 82: In reference to Tzedakis 2007, it would be helpful to expand on the differences between the Tzedakis study and the other studies in regard to the age of the onset of the Mediterranean climate regime as it is directly pertinent to this study. This would be especially helpful in regards to understanding the different criteria upon which these inferences were made.

Following the reviewer's suggestion, we elaborated about the differences between the Tzedakis study and the other studies mentioned. We indicated the different criteria on which each of them is based: 'the other studies' are chiefly based on pollen data, whereas Tzedakis' work is a review article that mentions several lines of evidence pointing towards an early existence of a Mediterranean climate. These lines of evidence are now enumerated in the manuscript.

Methods: Did you collect data on bone orientation? This would be especially useful and important for shoreline environments. Even in other environments, describing the orientations of bone long axes provides a context and process for disarticulation in different environments, especially during decomposition and other early-stage biostratinomic processes.

Unfortunately and given the already very long list of taphonomic variables recorded during field work, we did not measure orientation of the bones. We acknowledge that orientation can provide useful taphonomic information, above all, in habitats such as the beach or the river margin, so we will try to record this variable in future sampling.

The authors mention that they utilized Behrensmeyer's weathering scale based on Ambolseli work. As they correctly note, this scale is specific to the tropical savannah conditions of Kenya. While the authors mention this caveat, I encourage them to discuss more about the potential differences and modifications to this scale that may be a better tool for a Mediterranean study.

After giving a thorough consideration to this comment and reading more literature about weathering in different climatic regimes, we do not think that elaborating on this topic would shed more light than what is already explained in our paper: that a calibration of Behrensmeyer's weathering scale is needed in areas with a Mediterranean climate and that we are currently conducting our own weathering experiment at Doñana National Park to clarify this. Any other statement that we could add would be in the realm of speculation.

We have assembled the following information about weathering under different climatic regimes: 

Tappen (1994) studied bone weathering in a savannah environment at Parc National des Virungas, Secteur Nord (Democratic Republic of Congo), and found a similar calibration to that of Behrensmeyer (1978) at Amboseli. Nevertheless, Tappen also studied weathering at Ituri Rainforest (Democratic Republic of Congo) and observed that bone weathering was delayed or even absent.

In desert conditions (Jebel Barakah, UAE), Andrews (1995) reported weathering rates similar to those in Amboseli. 

In temperate environments, such as Yellowstone (Wyoming, USA) or Wales (UK), authors have reported extended weathering durations compared to those of Amboseli (Andrews 1995, Miller 2011).

What to expect in Mediterranean ecosystems? We do not yet know, probably an intermediate situation between tropical and temperate climates, but this is just a hunch and we prefer to wait until we can provide information from our experiment. 

Andrews, P. Experiments in taphonomy. Journal of Archaeological Science. 1995; 22: 147–153.

Behrensmeyer AK. Taphonomic and ecologic information from bone weathering. Paleobiology. 1978; 4: 150–162.

Miller JH. Ghosts of Yellowstone: multi-decadal histories of wildlife populations captured by bones on a modern landscape. PLOS ONE. 2011; 6: e18057.

Tappen, M. Bone weathering in the tropical rain forest. Journal of Archaeological Science. 1994; 21: 667–673.

---

## [Editor Report · Decision Letter 1]

27 Oct 2020

Taphonomic information from the modern vertebrate death assemblage of Doñana National Park, Spain

PONE-D-20-23069R1

Dear Dr. Soledad Domingo,

We’re pleased to inform you that your manuscript has been judged scientifically suitable for publication and will be formally accepted for publication once it meets all outstanding technical requirements.

Kind regards,

Luca Pandolfi

Academic Editor

PLOS ONE

Additional Editor Comments (optional):

Dear Authors,

thank you for your revised version.

The reviewers' suggestions have been satisfied and your replies are resonable and well-explained.

Sincerely,

Luca Pandolfi
---

## [Editor Report · Acceptance letter]

29 Oct 2020

PONE-D-20-23069R1 

Taphonomic information from the modern vertebrate death assemblage of Doñana National Park, Spain 

Dear Dr. Domingo:

I'm pleased to inform you that your manuscript has been deemed suitable for publication in PLOS ONE. Congratulations! Your manuscript is now with our production department. 

Kind regards, 

on behalf of

Dr. Luca Pandolfi 

Academic Editor

PLOS ONE